# Peripheral Biomarkers in Manifest and Premanifest Huntington’s Disease

**DOI:** 10.3390/ijms24076051

**Published:** 2023-03-23

**Authors:** Emanuele Morena, Carmela Romano, Martina Marconi, Selene Diamant, Maria Chiara Buscarinu, Gianmarco Bellucci, Silvia Romano, Daniela Scarabino, Marco Salvetti, Giovanni Ristori

**Affiliations:** 1Department of Neurosciences, Mental Health and Sensory Organs (NESMOS), Sant’Andrea Hospital, Sapienza University of Rome, 00189 Rome, Italy; emanuele.morena@uniroma1.it (E.M.);; 2Department of Human Neurosciences, Sant’Andrea Hospital, Sapienza University of Rome, 00189 Rome, Italy; 3Institute of Molecular Biology and Pathology, National Research Council, 00185 Rome, Italy; 4IRCCS Istituto Neurologico Mediterraneo (INM) Neuromed, 86077 Pozzilli, Italy; 5Neuroimmunology Unit, IRCCS Fondazione Santa Lucia, 00179 Rome, Italy

**Keywords:** Huntington’s disease, peripheral biomarker, plasma, blood, biomarker, premanifest, manifest, gene therapy, mHTT, neurofilament light chain, DNA damage response, leukocyte telomere length

## Abstract

Huntington’s disease (HD) is characterized by clinical motor impairment (e.g., involuntary movements, poor coordination, parkinsonism), cognitive deficits, and psychiatric symptoms. An inhered expansion of the CAG triplet in the huntingtin gene causing a pathogenic gain-of-function of the mutant huntingtin (*mHTT*) protein has been identified. In this review, we focus on known biomarkers (e.g., mHTT, neurofilament light chains) and on new biofluid biomarkers that can be quantified in plasma or peripheral blood mononuclear cells from mHTT carriers. Circulating biomarkers may fill current unmet needs in HD management: better stratification of patients amenable to etiologic treatment; the initiation of preventive treatment in premanifest HD; and the identification of peripheral pathogenic central nervous system cascades.

## 1. Introduction

Huntington’s disease (HD) is an autosomal-dominant inherited, neurodegenerative disorder caused by a cytosine–adenine–guanine (CAG) polyglutamine repeat expansion in the first exon of the HTT gene encoding the huntingtin protein [1]. The mutant protein (mHTT) contains an expanded polyglutamine sequence (poly Q) that confers a toxic gain-of-function [2], and CAG triplets ≥40 repeats result in a fully penetrant manifestation of the disorder [3]. HD is characterized by signs and symptoms of motor impairment (e.g., poor coordination, involuntary movements, altered eye movements), and cognitive and psychiatric symptoms. Onset occurs typically in adulthood (around 45 years of age) but depends on CAG repeat length. Symptoms progress irreversibly until death [4]. Manifest HD (mHD) is diagnosed when unequivocal extrapyramidal signs (e.g., chorea, dystonia, bradykinesia, rigidity) are present, with no explanation other than HD [5]. The diagnosis is established by genetic testing to determine the CAG repeat length in the HD gene [6].

## 2. Perimanifest, Premanifest, and Manifest HD

Huntington’s disease is diagnosed according to the diagnostic confidence level of the Unified Huntington’s Disease rating scale (UHDRS) [7], a clinical scale for evaluating motor, cognitive, behavioral, emotional, and functional signs and symptoms separately [8]. Individuals carrying the HD gene mutation but not displaying motor symptoms or signs are categorized as having premanifest HD (preHD), while those with slight symptoms but not yet manifesting unequivocal signs are categorized as having perimanifest HD (periHD) or being in a prodromal stage of HD. PreHD extends from birth to the beginning of the prodromal stage [9]. 

Previous studies investigating differences between preHD and healthy controls found slight changes in handwriting movements [10], gait, and posture [11] in preHD, in addition to autonomic dysfunction such as difficulty swallowing and lightheadedness on standing up [12]. 

Neuropsychiatric symptoms are a cardinal feature of HD: apathy is often noted to develop in preHD and periHD [13]. Finally, a prospective observational study (TRACK-HD) showed that baseline imaging and quantitative motor and cognitive measures have prognostic value in preHD to predict subsequent clinical diagnosis, independent of age and CAG repeat length (Figure 1) [14].

## 3. Informative Biomarkers

Gene therapy relies on informative biomarkers for identifying disease progression and staging a therapeutic response. A wide panel of biomarkers for HD (Figure 2) may become candidates for designing preventive treatment of preHD [15]. Current disease-modifying treatment for HD aims to lower HTT gene expression in the brain, thus delaying the development of pathogenic pathways due to the mutant protein [16]. A recent phase IIa/III trial testing *tominersen* (formerly known as IONIS-HTTRx and *RG6042*), an investigational antisense medicine designed to reduce HTT production, failed in a large cohort of patients with early-stage manifest HD (mHD), despite promising results obtained in phase II [17]. Analysis by an independent data monitoring committee evaluating the data from 60% of patients at 69 weeks into the study revealed clinical worsening in the patients receiving the study drug compared to those in the placebo arm. Worsening was greater in the HD patients in the arm treated every 8 weeks than in those treated every 16 weeks [18]. The evaluation tools in the trial (clinical scales, UHDRS, mHTT dosage in joint-to-volume magnetic resonance imaging (MRI) and cerebrospinal fluid (CSF)) may explain the cause for the failure. More accessible and reliable indicators of disease severity, progression, and phenotype in the early phase of mHD or the peri and the preHD phase could inform the design of disease-modifying approaches in future trials [19]. 

Numerous potential HD biomarker candidates have been investigated: biofluid biomarkers from CSF and plasma biomarkers; structural, functional, or biochemical imaging; electrophysiologic measures; quantitative clinical measures; digital biomarkers [20]; neuroimaging; and complementary biofluid biomarkers [21]. For instance, peripheral biofluid biomarkers have gained increasing attention because they can be quantified in body fluids with minimal invasiveness, good accuracy, and high discriminatory power. Recent studies have evaluated their role as biomarkers in HD [22,23]. 

Here, we conducted a literature review with a special focus on the characteristics of HD biomarkers, their biological relationships, and the differences between so-called “reporters” and standardized serum, plasma, and peripheral blood cell biomarkers. Data on peripheral biomarkers and their clinical correlations in the pre/perimanifest phase may advance our understanding of HD and how to measure treatment efficacy in its early phases.

## 4. Fluid Biomarkers: Peripheral Assessment

Peripheral biomarkers are derived from various body fluids, including blood, saliva, urine, and CSF. In neurodegenerative diseases, CSF provides the best source of biomarkers because it is enriched in CNS-derived substances. However, CSF is collected by lumbar puncture, an invasive procedure not routinely performed for HD diagnosis. Instead, the patient’s blood provides a more easily, accessible, and cost-effective fluid for biomarker evaluation.

Extensive studies on fluid biomarkers, especially mHTT and neurofilament light chain (NfL), have quantified CSF levels in blood and in peripheral blood (mononuclear cells and plasma). In the next paragraphs, we survey investigations on those quantified in both CSF and blood, as well as recent reports on the tubulin-associated unit (TAU) protein in CSF. Then we focus on biomarkers from peripheral blood (mononuclear cells and plasma): DNA damage response elements, telomere length (LTL) in white blood cells (leukocytes), and circulating non-coding RNA (NcRNA), which can represent peripheral reporters of CNS damage evaluated in blood.

### 4.1. Mutant Huntingtin (mHTT) 

mHTT plays a central role in the pathogenesis of HD, making it a key potential biomarker of interest, notwithstanding certain limitations. mHTT is present in low concentration in easily accessible biofluids but is difficult to distinguish between CNS-derived and peripheral mHTT, given that it is produced ubiquitously [19]. 

Weiss and colleagues in 2009 quantified mHTT in human whole blood, isolated erythrocytes, and buffy coats [24]. Clinical studies on soluble mHTT in blood showed contrasting results, however. In 2012, a time-resolved Förster resonance energy transfer (TR-FRET) immunoassay to quantify mutant and total HTT protein levels in leukocytes revealed insignificant differences in total HTT between patients with HD and healthy controls. However, the mean mHTT levels seemed to discriminate preHD vs mHD and preHD vs early-stage mHD without significant differences between the clinical stages of mHD [25]. A 2013 study using a homogeneous time-resolved fluorescence (HTRF) assay showed similar contrasting results for mHTT in leukocytes: the progressive and inverse relationship between decreasing mHTT levels and disease progression suggested that HD phenoconversion could be tracked. Owing to technical issues, however, the assay was not deemed ready for wider use [26]. Later, in 2015, Wild et al. demonstrated that mHTT can be quantified in CSF with a femtomolar-sensitive single molecule counting (SMC) immunoassay [27]. With this novel technique, the researchers detected no mHTT signal in the healthy controls. The assay demonstrated clinical sensitivity, specificity, and intraindividual stability over time in the CSF of patients with HD. The mHTT level in the CSF was correlated with the probability of disease onset, independent of known predictors of premanifest disease, such as age and CAG repeat length. mHTT levels correlated with disease severity in patients with mHD, as well as with CSF tau and NfLs, albeit a weaker clinical correlation than with NfLs [19]. The mHTT level in CSF was used as a biomarker of treatment effectiveness in an initial study based on Huntington-reducing therapy.

Despite its obvious role as the cause of HD, mHTT alone may not be the most useful single biomarker of clinical state or disease progression. Furthermore, owing to the difficulty of measuring peripheral mHTT and its ambiguous correlation with clinical signs and symptoms, measurement of mHTT (primary pathogenic agent) levels would need to be combined with other biomarkers, as well as NfLs (markers of axonal damage) to monitor disease progression and therapeutic effects [19].

Extracellular vesicles (EVs) have gained recent interest. EVs are phospholipid bilayer membranes that envelope particles and are released by the cells into the extracellular environment and body fluids. Depending on their biogenesis, three main subtypes are distinguished: exosomes, microvesicles, and apoptotic bodies [28,29]. EVs contain a wide range of bioactive molecules, including proteins, lipids, messenger RNA, micro RNA, lncRNA, and metabolites [30,31,32]. Small in size (40–120 nm), exosomes are released by nearly all cell types [33]. Because they have a key role in cell–cell communication [34], exosomes have potential as non-invasive diagnostic biomarker carriers for neurodegenerative diseases [35]. In addition, by virtue of their ability to cross the blood–brain barrier [36], they showed promise as therapeutic drug carriers in preclinical trials [37,38]. A recent study on an experimental model of HD and plasma from patients heralded the use of exosomes as biomarker carriers in HD: the HTT protein was co-isolated with EVs from the pig model and HD patient plasma; total huntingtin levels in the EVs were higher in the plasma of HD patients than healthy controls in pig models and HD patients [35].

### 4.2. Neurofilament Light Chain 

Neurofilament light chain (NfL) is a protein of the axonal cytoskeleton. While a reliable reporter of axonal damage, it is not disease-specific. Elevated NfL levels have been reported across a spectrum of neurological conditions, including HD [39].

NfLs were detected in the CSF of mHD patients by Costantinescu et al. in 2009 [40]. Later, Rodrigues and colleagues compared the longitudinal dynamics of HD biomarkers and found that CSF and plasma NfLs may perform better than mHTT: NfLs showed stronger correlation with non-biofluid measures (clinical, neuroimaging) than CSF mHTT in longitudinal analysis, proving its usefulness as a biomarker of disease status, clinical progression, and brain atrophy. It was also found that the concentration dynamics of NfLs in plasma and in CSF nearly overlapped [41]. 

Plasma NfLs were found to be a predictor of disease onset in preHD. A recent longitudinal study involving 112 patients showed a strong correlation between plasma NfL concentration and symptom onset, as assessed with the normalized prognostic index (PIN) [42]. The PIN score is a composite score that includes age, CAG repeat length, and clinical assessment, which may provide an accurate index of disease progression [43]. A cut-off of plasma NfLs (45.0 pg/mL) was found to distinguish the predicted disease onset between less and more than 10 years. Other studies showed the usefulness of plasma NfLs to distinguish preHD from mHD: NfL levels in plasma and CSF discriminated between healthy controls and patients with preHD, and between preHD and mHD, whereas mHTT levels correlated only between healthy controls and HD mutation carriers [19]. 

Moreover, compared to other measurements in neuroimaging studies, plasma NfL concentration evaluated in combination with quantitative magnetic resonance imaging (qMRI) showed that high plasma NfL levels correlated with microstructural degeneration of posterior cortical [44] and subcortical white matter [45]; however, no statistical correlations were found between plasma NfLs and caudate/putamen volume [45]. A correlation between plasma NfL concentration and clinical symptoms was found only when the cohort included preHD and mHD patients but not when the mHD and the preHD patients were evaluated separately. A recent retrospective study suggested a critical role of plasma NfLs in juvenile HD (JHD), a devastating and rare form of HD caused by exceptionally long CAG repeats that lead to motor manifestations before 21 years of age [46]. The plasma NfL levels were higher in the patients with JHD than in the healthy controls and in the preHD children. Moreover, there was a strong correlation between plasma NfL levels in patients with JHD and caudate and putamen volumes [47]. However, the results in adult HD forms left unanswered the question of the usefulness of plasma NfLs to track disease severity and progression, thus potentially limiting its use in clinical trials [48].

### 4.3. Tubulin-Associated Unit (TAU) Protein 

TAU is a microtubule-associated protein that promotes their assembly and stability in the axons of the CNS [49]. Altered tau biology has been reported in Alzheimer’s disease, Parkinson disease, and other dementia-related neurodegenerative disorders [50]. In 2015, tau was quantified by enzyme-linked immunosorbent assay (ELISA) for the first time in the CSF of patients with HD: the CSF tau level distinguished between the healthy controls and the carriers of HD gene expansion (premanifest and manifest evaluated together). Clinical features, and motor test findings in particular, correlated with CSF tau concentration [51]. Direct comparison of CSF levels showed better correlation between clinical tests and NfLs than with tau protein [52].

Post-mortem brain analysis of patients with advanced clinical stage HD has revealed abnormal tau deposition and other features of tauopathies [53]. Evidence for correlations between hyperphosphorylated tau biology and HD [54,55] suggest HD as a secondary tauopathy and the opportunity of directly targeting tau [56].

Nonetheless, because CSF tau protein remains a high-cost, difficult-to-access biomarker, finding a reliable blood substrate for its measurement will be crucial in the treatment of HD, and of Alzheimer’s disease especially [57]. Finally, longitudinal analysis will need to establish the utility of tau in predicting disease progression, phenotypic variability, and therapeutic response.

### 4.4. DNA Damage Response in Peripheral Blood Leucocytes

The accumulation of DNA damage induced by biological insult in repair-defective individuals may lead to neuronal cell death by either progressively depriving the cell of vital transcripts or apoptosis [58,59]. DNA damage in eukaryotic cells consists mainly of single-strand breaks (SSBs) and double-strand breaks (DSBs). In response to unrepaired DNA damage, a signaling cascade involves various different proteins—DNA damage sensors, transducers, mediators, and effectors—that interact in damage DNA response (DDR). Emerging evidence has pinpointed a role for the modulation of chromatin organization in DDR [60,61], the most widely documented being phosphorylation of the histone variant pγ-H2AX, which was identified more than 10 years ago [62,63]. 

A recent collaborative study evaluated DDR in the peripheral blood mononuclear cells (PBMCs) from 58 patients with HD, 23 with preHD with a similar expansion in the HTT gene (43.4–44.9 CAG repeats), and 18 healthy controls [64]. Phosphorylated γ-H2AX (pγ-H2AX) fluorescence, which marks double-strand DNA breaks, was analyzed by cytofluorimetry isolated from fresh peripheral lymphocytes. DNA damage was greater in the PBMC from the patients with preHD and mHD compared to the healthy controls (HC vs mHD, *t*-test *p* = 0.000001 area under the receiver operating characteristic (ROC) curve (AUC) 0.87; HC vs preHD, *t*-test *p* = 0.0008, AUC 0.88). Four patients with preHD were analyzed at several time points: the pγ-H2AX signal increased over time, although clinical manifestations were absent, and the pγ-H2AX signal decreased after the onset of clinical signs in three patients but remained higher compared to the healthy controls. In the patients with HD and preHD, the pγ-H2AX levels correlated with progression of the HD phenotype measured 3 years later. Progressive DNA damage in lymphocytes without the manifestation of clinical signs was noted in the patients with preHD. Basal levels of DNA damage were found to predict faster progression of disease. With onset of the clinical manifestations, the DNA damage signatures stabilized at higher levels than in the healthy controls over time. These features, together with negligible DDR damage in the healthy controls and potential reversibility of the DDR cascade, suggest the use of peripheral blood lymphocytes as quick and handy biomarkers to determine disease progression and response to treatment in HD.

### 4.5. Leukocyte Telomere Length

Telomeres are the terminal ends of chromosomes and play a role in preserving genome stability. Telomere shortening occurs progressively with repeated cell division because of the inability of DNA polymerase to replicate the 3′ end of the DNA strand. Telomerase, a cellular multiprotein complex, counteracts telomere shortening. While present in the early stages of embryonic development, its activity is silenced in certain human somatic tissues immediately after birth. As a result, telomeres shorten progressively with age in the replicating cells of adult tissues [65]. This phenomenon may indicate cellular senescence and reflect an organism’s biological age. Peripheral blood mononuclear cells (PBMCs) are ideal for telomere research for several reasons: they are easy to obtain from blood and are readily available, and since they circulate throughout the body, immune cells are exposed to both internal (from cells) and external (from diet and exposure) factors that affect telomere maintenance. 

Shortened leukocyte telomere length (LTL) has been found to be associated with cardiovascular diseases [66], diabetes, and metabolic syndrome [67], psychological disorders [68], auto-immune diseases [69], and Alzheimer’s disease [70,71]. A growing body of evidence suggests that oxidative stress (OxS) and chronic inflammation can contribute to telomeric attrition and the development of age-related diseases [72,73]. When LTL was investigated in neurodegenerative diseases, a shorter LTL was frequently found in association with cognitive decline/dementia and AD [70,71] and multiple sclerosis [69]. Progressive leukocyte telomere reduction was detected in patients with mild cognitive impairment (MCI) and AD compared to healthy controls [71], suggesting that LTL measurement could be a useful means to follow dementia progression as it converts from prodromal (MCI) to manifest AD. 

A key event in the pathogenesis of AD and HD is neuroinflammatory processes characterized by microglia activation and reactive astrocytes that cause transcriptional activation of pro-inflammatory genes that perpetuate a chronic inflammatory state [71,74,75]. The neuroinflammatory state accompanying HD and the concurrent involvement of the peripheral immune system may promote leukocyte division and telomere shortening in relation to astrocyte and microglia proliferation [70]. Considerably shorter LTL has been observed in HD patients than in age-matched healthy controls [64,76,77,78]. A study investigating LTL in a cohort of 38 patients with preHD, 62 with HD, and 76 healthy controls observed significant differences in LTL between the three groups (*p* < 0.0001) in the order HD < preHD < healthy controls [77]. The mean LTL in the patients with mHD was about half that of the healthy controls and slightly greater than the minimum length reported to be necessary to ensure human telomere protective stability in leukocytes [65]. The overall data seem to indicate that LTL begins to shorten markedly in patients with preHD to lengths observed in patients with symptomatic HD. Furthermore, the linear relationship between LTL and estimated years to clinical onset of HD was found to reliably predict the time at clinical diagnosis. A recent follow-up study involving patients with preHD carrying alleles with different CAG repeat numbers was conducted to identify a threshold LTL close to the time of disease onset. A marked reduction in telomere length in the patients with preHD was noted about 2.5 years before HD onset compared to the healthy controls and independent of CAG size. This homogeneity allows for a common cut-off of 0.70 T/S (number of copies of telomeric repeats (T) compared to a single-copy gene (S) used as a quantitative control) to distinguish between preHD and mHD. LTL > 0.70 may indicate a premanifest stage at about 3 years before clinical onset, while LTL < 0.70 T/S may indicate imminent clinical onset [79]. Taken together, the data indicate that LTL may possess required characteristics of an ideal biomarker of HD progression [23]. It can be easily obtained by inexpensive blood sampling, it is readily quantifiable and highly reproducible, and is closely linked to the pathophysiology of HD.

### 4.6. Non-Coding RNAs (ncRNAs)

The mammalian genome contains sequences for proteins encoding RNA, called messenger RNA (mRNA) and non-coding RNA (ncRNA). The catalog of known ncRNAs includes, among others, long non-coding RNAs (lncRNAs; longer than 200 nucleotides), circular RNAs (circRNAs; generated from pre-mRNA backsplicing), small non-coding microRNAs (miRNAs; around 21–25 nucleotides), and natural antisense transcripts (NATs) generated by transcription in the opposite direction to protein coding transcripts [80].

MicroRNAs are the most widespread subtype ncRNA. They are short (19–25 nucleotides long), regulatory RNA molecules that affect translation and stability of their mRNA targets by guiding the RNA-induced silencing complex (RISC) predominantly to the 3′ untranslated region (UTR) [81,82]. MicroRNAs also influence many aspects of metazoan biology primarily by mediating mRNA stability and preventing translation [83]. Changes in the level of circulating miRNAs have been associated with a wide range of diseases, including type 2 diabetes, obesity, cardiovascular disease, cancer, and neurodegenerative disorders. Transcriptional dysregulation has long been recognized as central to the pathogenesis of HD [84].

Post-mortem analysis of brain tissue has shown dysregulation of microRNAs in the cortex of patients with HD [85]. Circulating microRNAs may be detected in biofluid (CSF [86], PBMCs [87], and plasma [88]). Alteration in miRNAs, such as miR-9/9∗ [89], miR-168 132, miR-4488, miR-196a-5p, and miR-549a, and recently-miR-323b-3p [88,90,91,92] among others, has been found in post-mortem analysis, supporting the hypothesis for a crucial role of miRNA in HD [93].

To date, there is little information about the role of ncRNAs in preHD [94,95]. Studies of small nucleolar RNAs (snoRNA) and their potential as HD biomarkers and their role in HD pathophysiology are lacking. SnoRNAs (60–300 nucleotides in size) accumulate primarily in the nucleoli and regulate rRNA transcription. SnoRNAs are excised from the introns of pre-mRNAs, which also generate functional mRNAs from their exonic regions. Two classes are distinguished: C/D box snoRNAs and H/ACA box snoRNAs. C/D box snoRNAs guide 2′-O-ribose methylation, while H/ACA box snoRNAs direct pseudouridylation of nucleotides [96]. 

Our group recently investigated the role of plasma U13 snoRNA (SNORD13). With this cross-sectional study we found a higher plasma concentration of SNORD13 in 23 patients with mHD compared to 16 with preHD and to the control cohorts (24 psychiatric patients, 28 patients with Alzheimer’s disease, and 24 healthy subjects). We also found a correlation between SNORD13 and the status of mutant huntingtin carriers and HD disease but not the CAG number [97]. In a future longitudinal study with a larger sample, SNORD13 will be investigated as a potential HD-specific biomarker and indicator of new pathways amenable to treatment.

## 5. Conclusions

Numerous efforts are under way to find useful biomarkers for HD. In this review, we focused on peripheral fluid biomarkers in the pre/perimanifest phases of HD. HD biomarkers have been informative for their potential predictive role in monitoring disease course and therapeutic response. Each biomarker has certain limitations, however. NfLs are not specific for HD but rather are the expression of irreversible, general neuronal damage. Tau protein, as well as being non-specific for HD, is not a useful biomarker because it is difficult to obtain and its role in HD pathology remains to be elucidated. While CAG expansion in the HTT gene is a known unequivocal cause of HD, mHTT varies considerably with age of onset and clinical disease progression, suggesting that some genetic and environmental modifiers of disease may impact the effects of mHTT. Furthermore, the difficulty of measuring mHTT in peripheral blood and the lack of consistent correlations with clinical assessment of mHD preclude the use of peripheral mHTT as a biomarker for HD. Differently, NcRNAs are highly stable in biological fluids and may mediate paracrine and endocrine effects on different tissues, modulate gene expression, and the function of distal cells [98]. These features support the notion of NcRNAs as possible modifiers of HD. A clinical trial investigating a specific microRNA, via a viral delivery approach (rAAV5-miHTT), is currently under way with the aim to inhibit mHTT production (NCT04120493) [99]. However, we still know very little about ncRNAs and studies performed so far have produced inconsistent results. NcRNAs, LTL, and DDR are understudied. Their cumulative role, along with known biomarkers, may open a window on the complex pathogenic scenario of HD. A future area of focus should be peripheral biomarkers and assessment of their ability to predict disease progression and treatment efficacy in preHD.

## Figures and Tables

**Figure 1 ijms-24-06051-f001:**
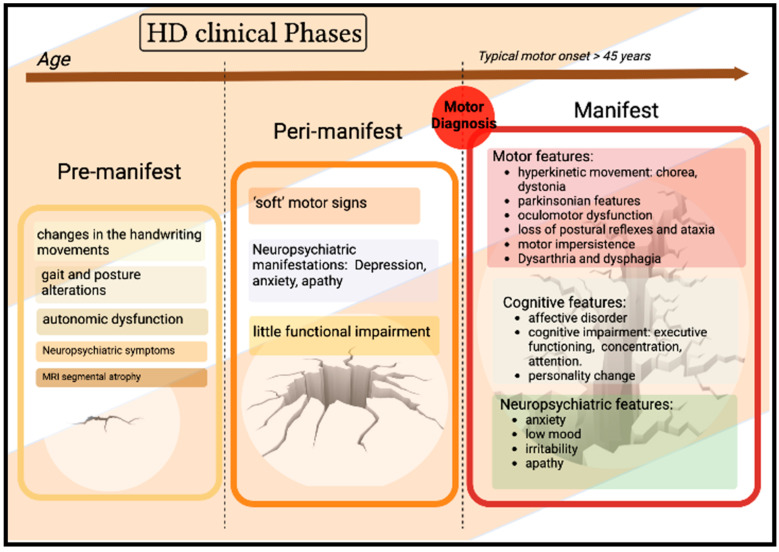
Clinical phases of Huntington’s disease [9].

**Figure 2 ijms-24-06051-f002:**
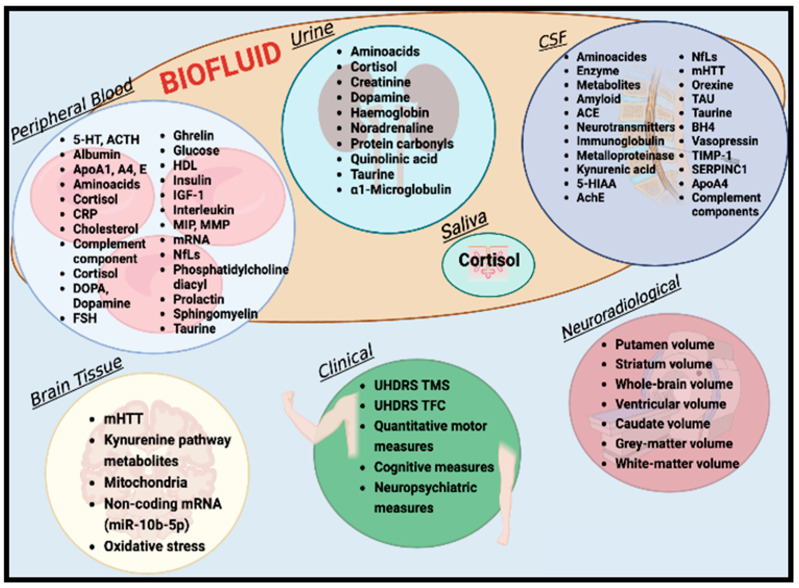
Biomarkers for Huntington’s disease [23].

## Data Availability

Not applicable.

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
