# Peer review of "Peripheral Biomarkers in Manifest and Premanifest Huntington’s Disease"

_ijms, 2023, doi:10.3390/ijms24076051_

Round 1

Reviewer 1 Report

This manuscript is a review of biomarkers for Huntington’s disease (HD) progression and symptom onset prediction. The authors make the case that there are unexplored minimally invasive biomarkers which show promise for being specific to HD including noncoding RNAs, leukocyte telomere length, and DNA damage response. However, although the paper is an interesting nice paper, it is hard to understand what is being said in many places due to what is probably English translation. The paper requires extra work to correct all the missing words, mixed up grammar, spelling, and punctuation.

Point 1:  The English grammar, punctuation and spelling are not up to journal standards. Please correct all spelling, punctuation and grammar errors. For some examples see lines 18, 19, 21, 22, 25, 26, 27, 27, 30, 31, 32, 47, 60, 64, 70, 73, 74, 76, 78, 79, 85, 88, 103, 114, 126, 133, 168, 177, 194, 216, 227, 230, 232, 233, 234, 277, 332, 333, 334, 335, 351, 361, 362, 366, 369, 370, 373, 374, and 377.  Also please look at figure 1 ‘presymptomaic’, ‘parkinsonia’?, ‘dysathria’?, and figure 2 ‘amminoacid’?, ‘phospatidyicholine’?, ‘neurotrasnmitter’?.

Point 2:  In the heading for section (2.), should it be ordered as “Characteristics of premanifest, perimanifest, and manifest phases.”  ??

Point 3:  In line 30, Huntington’s should be capital H.

Point 4:  In line 127, should this statement be referenced to which study quantified these types of RNAs?

Point 5:  In line 177, should ext. be ect.?

Point 6:  In line 308, T/S requires a definition.

Point 7:  In lines 337-338, this sentence makes no sense and seems to be key to the whole paper. Please reword.

Point 8:  In line 376, is it possible to define “approach” rather than bundling future plans into this one broad word?

Point 9:  The final sentence of the paper (lines 378-380) is key to the importance of the review but unfortunately starting with ‘Especially’, confuses the meaning of the last sentence.  This sentence needs rewording to drive the point home.

Author Response

Point 1:  The English grammar, punctuation and spelling are not up to journal standards. Please correct all spelling, punctuation and grammar errors. For some examples see lines 18, 19, 21, 22, 25, 26, 27, 27, 30, 31, 32, 47, 60, 64, 70, 73, 74, 76, 78, 79, 85, 88, 103, 114, 126, 133, 168, 177, 194, 216, 227, 230, 232, 233, 234, 277, 332, 333, 334, 335, 351, 361, 362, 366, 369, 370, 373, 374, and 377.  Also please look at figure 1 ‘presymptomaic’, ‘parkinsonia’?, ‘dysathria’?, and figure 2 ‘amminoacid’?, ‘phospatidyicholine’?, ‘neurotrasnmitter’?

Answer: We corrected the spelling mistakes in figure 1 and 2. We send the manuscript to an English reviewer to ameliorate spelling and grammar form.

Point 2:  In the heading for section (2.), should it be ordered as “Characteristics of premanifest, perimanifest, and manifest phases.”  ??

Answer: We agree with Point 2 and we made adjustments.

Point 3:  In line 30, Huntington’s should be capital H.

Answer: We made the change proposed in Point 3.

Point 4:  In line 127, should this statement be referenced to which study quantified these types of RNAs?

Answer: We reworded the sense of sentence in line 127 (Point 4), without the need to a reference.

Point 5:  In line 177, should ext. be ect.?

Answer: Point 5, we corrected ext. with etc. (et cetera, correct adverb spelling take from Cambridge Dictionary).

Point 6:  In line 308, T/S requires a definition.

Answer: We explained the meaning of T/S in line 308 (in new version 357)

Point 7:  In lines 337-338, this sentence makes no sense and seems to be key to the whole paper. Please reword.

Answer: We reformulated the sentence in 337-338 as follows: To date, there is little information about the role of ncRNAs in preHD [95], [96]. (now in the lines 387).

Point 8:  In line 376, is it possible to define “approach” rather than bundling future plans into this one broad word?

Answer: We modified all the paragraph.

Point 9:  The final sentence of the paper (lines 378-380) is key to the importance of the review but unfortunately starting with ‘Especially’, confuses the meaning of the last sentence.  This sentence needs rewording to drive the point home.

Answer: We agree with Point 9, the crucial role of this final paragraph. We modified the sentences, making them shorter and directed to review’s aim.

Reviewer 2 Report

Dear Authors, 

Your work summarises the advances in the research of biomarkers for HD. However, it is not yet clear which biomarkers you think might be useful. The text seems to me to be out of order jumping from studies that have been done in plasma to others that have been done in CSF. I would suggest that you order it so that the reader can follow the text. I recommend that you make a short summary of the pros and cons after describing each biomarker. 

I would like to note some revisions:

Abstract, line 20:  therapy, with (no coma), line 22, evidence 

Introduction, line 37:  The mutant protein 37 (mHTT), contains (no coma). 

Line  47: HD.[5]  First the reference after the dot. During the text, there are more errors of this type.

Line 50: you write: (...) motor signs and the other symptoms and then in lines 52-53 you describe them. I recommend that you write it all in the same sentence.

Lines 69-71: I would rewrite the sentence. Take out MRI tools and put the reference after figure 1 and dot. 

Line 77: Huntington’s disease  HD.

Lines 87 and 90: put them into the same paragraph that starts in line 77. 

Line 97: remove complementary biofluid biomarker.

Reference style: [23, 24].

Line 118: I would remove that whole paragraph and put in an introductory sentence for the biomarkers you describe next.

mHTT: the objective of the review is to focus on blood biomarkers and the differences between the different phases and here you do not talk about this at all. 

Line 210: You have already described the acronym in the section title, I don't think you should repeat it, let alone put a reference to it. In this same section, you do not talk about blood. 

Conclusion: You should focus on the blood biomarkers that you think maybe most useful and forget about the pros and cons in this section and not put references. 

Figures 1-2: There are sentences that begin with capital letters and others with lowercase letters. Put all of them in capital letters.

Best regards, 

Author Response

Your work summarises the advances in the research of biomarkers for HD. However, it is not yet clear which biomarkers you think might be useful. The text seems to me to be out of order jumping from studies that have been done in plasma to others that have been done in CSF. I would suggest that you order it so that the reader can follow the text.

Answer: We understand reviewer’s point of view. See the first part of section 4, it was rephrased to fulfill this reviewer’s suggestion and explain better the organization of the following paragraphs. 

I recommend that you make a short summary of the pros and cons after describing each biomarker.

Answer: See below “conclusions” and the following answer.

I would like to note some revisions:Abstract, line 20:  therapy, with (no coma), line 22, evidence.

Answer: We removed comma from lines 20 of the abstract and put evidence at the singular.

Introduction, line 37:  The mutant protein 37 (mHTT), contains (no coma).

Answer: We deleted comma in line 37

Line  47: HD.[5]  First the reference after the dot. During the text, there are more errors of this type.

Answer: We put all the reference before the dot (5, 76, 88); we checked there is always space between last word and the reference. Moreover, we sent the manuscript to an independent English reviewer to control and correct grammar and spelling mistakes.

Line 50: you write: (...) motor signs and the other symptoms and then in lines 52-53 you describe them. I recommend that you write it all in the same sentence.

Answer: Line 50 (now 50) we reworded the sentences as suggested.

Lines 69-71: I would rewrite the sentence. Take out MRI tools and put the reference after figure 1 and dot.

Answer: we modified the entire paragraph .

Line 77: Huntington’s disease  HD.

Answer: Line 77, HD was switched (now line 67).

Lines 87 and 90: put them into the same paragraph that starts in line 77.

Answer: The three paragraphs were combined together.

Line 97: remove complementary biofluid biomarker.

Answer: At line 97 we accepted your suggestion and we removed the sentence: “..and complementary biofluid biomarker”.

Reference style: [23, 24].

Answer: we change in the citation style required by the journal

Line 118: I would remove that whole paragraph and put in an introductory sentence for the biomarkers you describe next.

Answer: Line 118 (now 104), we modified all the paragraph.

mHTT: the objective of the review is to focus on blood biomarkers and the differences between the different phases and here you do not talk about this at all.

Answer: We agree with reviewer’s point of view. We will cite in the text more studies about peripheral mHTT biomarkers and we will reworded the paragraph to highlight the evidences about mHTT assessment in blood and its correlation with premanifest HD, as well as its clinical measurement in manifest HD

Line 210: You have already described the acronym in the section title, I don't think you should repeat it, let alone put a reference to it. In this same section, you do not talk about blood.

Answer: We deleted the acronym. We describe at the beginning of paragraph 4 that we will talk about different peripheral biomarkers, someone in both CSF and blood, someone else only in the blood. Moreover, there is someone else, as well as TAU, that will be useful in blood in the next future. For this reason, we think that is important to talk of all these biomarkers.

Conclusion: You should focus on the blood biomarkers that you think maybe most useful and forget about the pros and cons in this section and not put references.

Answer: We disagree with the reviewer’s suggestion. In our opinion the topic of biomarkers in HD is still in progress, and we do not feel in the position of prioritize some of them. We therefore prefer to maintain the current structure of the conclusion (including the pros and cons of the discussed biomarkers) and just to indicate possible future directions.

Figures 1-2: There are sentences that begin with capital letters and others with lowercase letters. Put all of them in capital letters.

Answer: We put all the sentences and words began with capital letter.

Round 2

Reviewer 1 Report

spell check required - is amino acid spelled amminoacid?; also I saw in the middle of sentences ncRNA as NcRNA.

Other than minor spelling, I am satisfied with the revision. Thanks.

Reviewer 2 Report

Dear authors, 

Thank you very much for the work done.  Remember that in the figure neuropsychiatric and depression are still capitalized.

Best regards,